# Stress-Inducible Gene *Atf3* Dictates a Dichotomous Macrophage Activity in Chemotherapy-Enhanced Lung Colonization

**DOI:** 10.3390/ijms22147356

**Published:** 2021-07-08

**Authors:** Justin D. Middleton, Jared Fehlman, Subhakeertana Sivakumar, Daniel G. Stover, Tsonwin Hai

**Affiliations:** 1Department of Biological Chemistry and Pharmacology, College of Medicine, Ohio State University, Columbus, OH 43210, USA; middleton.124@osu.edu (J.D.M.); jfehlman@neomed.edu (J.F.); sivakumar.35@buckeyemail.osu.edu (S.S.); 2Molecular, Cellular and Developmental Biology Program, Ohio State University, Columbus, OH 43210, USA; 3Department of Internal Medicine, College of Medicine, Ohio State University, Columbus, OH 43210, USA; Daniel.Stover@osumc.edu

**Keywords:** chemotherapy, breast cancer metastasis, lung colonization, macrophage dichotomy, stress response, *Atf3*

## Abstract

Previously, we showed that chemotherapy paradoxically exacerbated cancer cell colonization at the secondary site in a manner dependent on *Atf3*, a stress-inducible gene, in the non-cancer host cells. Here, we present evidence that this phenotype is established at an early stage of colonization within days of cancer cell arrival. Using mouse breast cancer models, we showed that, in the wild-type (WT) lung, cyclophosphamide (CTX) increased the ability of the lung to retain cancer cells in the vascular bed. Although CTX did not change the WT lung to affect cancer cell extravasation or proliferation, it changed the lung macrophage to be pro-cancer, protecting cancer cells from death. This, combined with the initial increase in cell retention, resulted in higher lung colonization in CTX-treated than control-treated mice. In the *Atf3* knockout (KO) lung, CTX also increased the ability of lung to retain cancer cells. However, the CTX-treated KO macrophage was highly cytotoxic to cancer cells, resulting in no increase in lung colonization—despite the initial increase in cell retention. In summary, the status of *Atf3* dictates the dichotomous activity of macrophage: pro-cancer for CTX-treated WT macrophage but anti-cancer for the KO counterpart. This dichotomy provides a mechanistic explanation for CTX to exacerbate lung colonization in the WT but not *Atf3* KO lung.

## 1. Introduction

Chemotherapy is a double-edged sword. It is cytotoxic to cancer cells but can also paradoxically promote cancer progression (reviews, [1,2,3,4,5,6,7]). Since chemotherapy is widely used to treat cancer, this is an important issue. One mechanism by which chemotherapy can bring about its paradoxical effect is to directly affect the cancer cells [8,9,10]. However, emerging literature indicates that changing the environment in the host, that is, the non-cancer cells, is an important mechanism (reviews above). A number of reports indicated that chemotherapeutic agents such as paclitaxel (PTX), doxorubicin, and gemcitabine increase the abundance of macrophages in the primary tumors. These macrophages exhibit pro-cancer activities, thereby enhancing tumor growth. The net effect is counteracting the efficacy of chemotherapy to reduce tumor size ([11,12,13,14]; reviewed in [2,3,4]).

In addition to enhancing tumor growth, chemotherapy can enhance metastasis by changing the non-cancer cells ([15,16,17,18,19,20], reviewed in [6,21]). At the primary tumor, PTX was shown to increase the abundance of a micro-anatomical structure called the tumor microenvironment of metastasis (TMEM) [19,20], which is composed of a perivascular macrophage and a cancer cell in close contact at the blood vessels [22]. Because cancer cells enter the blood stream at TMEMs [23], by increasing the abundance of TMEMs, PTX increases circulating cancer cells [19,20]. To address whether chemotherapy increases the ability of circulating cancer cells to colonize the distant organs, researchers used the lung colonization model coupled with a pre-treatment design, in which mice were treated with chemotherapeutic agents several days before intravenous injection of cancer cells for lung colonization. This pre-treatment design allows the drugs to be cleared from mice, eliminating its direct killing of cancer cells. Thus, if a drug has any effect on cancer burden, it must be due to its effect on the host cells, which in turn affects the ability of cancer cells to colonize the lung. The intravenous injection of cancer cells affords two advantages: (i) it bypasses the need for cancer cells to escape primary tumors, thus avoiding the complication from differences in tumor size; (ii) it ensures enough cancer cells for analysis. We will refer to this model as the pre-treatment lung colonization model. Using this model, we and others demonstrated that chemotherapy changes the lung environment to be more cancer friendly. As such, mice pre-treated with chemotherapeutic agents cyclophosphamide (CTX), PTX, or cisplatin had higher lung colonization than those with control treatment (see [19] supplemental for CTX; [15,16,19] for PTX; [15] for cisplatin).

Intriguingly, this chemotherapy-enhanced lung colonization is dramatically reduced in knockout (KO) mice deficient in *Atf3* [19], a gene known to play an important role in cellular stress response [24]. Since the same cancer cells (with *Atf3*) are injected, it means that *Atf3* in non-cancer host cells—referred to as the host *Atf3* below—is necessary for chemotherapy to efficiently enhance lung colonization. *Atf3* encodes a member of the ATF/CREB family of transcription factors [25,26]. The role of *Atf3* in stress response is manifested at two levels. First, the *Atf3* gene is expressed at a low level in most cells but is greatly induced by a wide variety of stimuli (a review, [27]). Second, at the basal low level under unstressed conditions, the *Atf3* gene product (ATF3 protein) already binds to many of its target genomic sites, prompting the idea that ATF3 “bookmarks” the genome to allow rapid transcriptional response upon stress stimulation [28]. Thus, as a stress-inducible gene, *Atf3* in the host cells links chemotherapy (a stressor) to altered transcriptional programs and makes the host environment more cancer friendly.

Tissue colonization is a rate-limiting step in metastasis [29]—the major cause of cancer death [30]. Thus, it is imperative to understand how chemotherapy, the very treatment for cancer, enhances colonization. However, our current understanding is rudimentary, and the current study is aimed to address this gap. In this report, we present evidence that Atf3 dictates a dichotomous (yin–yang) switch of macrophage. In response to chemotherapy, the wild-type (WT) macrophages exhibit a pro-cancer activity, but the Atf3 KO macrophages an anti-cancer activity. Our data support the idea that the antimicrobial gene cathelicidin, which is differentially upregulated by CTX in the *Atf3* KO macrophages, is an important anti-cancer factor. The significance and implication of these findings will be discussed.

## 2. Results

### 2.1. The Host Atf3, Combined with Chemotherapy, Establishes a Cancer-Friendly Tissue Environment at Early Stages of Lung Colonization

As described previously, using the pre-treatment lung colonization model, we found that chemotherapy enhances the ability of breast cancer cells to colonize the lung in the WT mice but not the *Atf3* KO mice. To investigate when this pattern is established, we carried out a time course analysis (at 3, 9, 18, 36, and 72 h after cancer injection) using CTX (a frontline chemotherapeutic drug) pre-treatment, followed by intravenous injection of MVT-1 breast cancer cells into FVB/N mice (Figure 1a). The MVT-1 breast cancer cells were labeled with turbo green fluorescent protein (tGFP) to facilitate analysis. Each cluster of cells (one or more cells, within 10 µm of each other) is counted as one microcolony. Figure 1b shows that CTX increased the colony numbers at 3 h in both WT and *Atf3* KO lung. At this time point, most cancer cells have not extravasated yet ([31]; a review [32]). Therefore, the increase in cancer cell numbers by CTX must be due to the change in the lung that resulted in higher cancer cell retention (presumably due to increased interactions between cancer cells and the vascular wall and, thus, decreased cancer cell loss by blood flow). This initial increase in cancer cells by CTX was maintained at later time points in the WT lung but dwindled away in the KO lung, leading to a host-*Atf3*-dependent increase in colonies by CTX as shown at 72 h. Analysis of cell numbers within each colony showed that they started to increase at 36 h, indicating colony growth—more cell proliferation than death. Figure 1c shows a representative image. Quantitation of the data at 72 h showed that CTX increased colony growth in WT but not *Atf3* KO lung (Figure 1d). Taken together, CTX increased lung colonization in a host-*Atf3*-dependent manner. Note that cancer cells were injected 4 days after CTX pre-treatment (Figure 1a). We also tested a 2-day pre-treatment regimen and found similar results at 72 h (not shown). However, the exacerbation by CTX in the WT host was not as robust. Therefore, all subsequent experiments were carried out using the 4-day regimen.

We also tested Met1 breast cancer cells, which differ from MVT-1 in their oncogenic events: Met1 cells are derived from breast tumors driven by the oncogene polyoma middle T antigen (PyMT) in a transgenic mouse model, whereas MVT-1 are from breast tumors driven by Myc and VEGF. As shown in Appendix A (see Appendix A), CTX also increased Met1 lung colonies in a host-*Atf3*-dependent manner, albeit at a delayed time point. Instead of 72 h, the dependence on host *Atf3* became apparent at 120 h. We next tested the applicability of this observation using PTX, another frontline chemotherapeutic agent, on a different strain of mice (C57BL/6) injected with PyMT breast cancer cells derived from tumors driven by PyMT in C57BL/6 mice [33]. As shown in Appendix A, PTX also increased lung colonies in this strain of mice in a host-*Atf3*-dependent manner.

Pre-treatment is an accepted model to study how chemotherapy may affect the host, which in turn affects cancer cells [15,16,19]. A caveat of this model is that it is not used in clinical practice. To address this caveat, we used a “neoadjuvant coupled with a double injection model” (see [34] and Appendix A), where chemotherapy is administered in the presence of a primary tumor—a clinically relevant context, referred to as neoadjuvant therapy. In this model, mCherry-labeled cancer cells were injected at a mammary fat pad to form primary tumor, followed by chemotherapy to mimic neoadjuvant therapy. Four days after chemotherapy, tGFP-MVT1 cells were injected (i.v.), and the lung was analyzed at 72 h later (Appendix A). The second injection of cancer cells by i.v. was necessary to ensure a sufficient number of cells for analysis. Appendix A shows that both PTX and CTX enhanced lung colonization of tGFP cells in a host-ATF3-facilitated manner, a pattern similar to that in Figure 1b and lending credence to the pre-treatment lung colonization model. We did not detect any primary tumor-derived cancer cells (mCherry-labeled), highlighting the need for i.v. injection of tGFP-labeled cells.

Despite using neoadjuvant to mimic a clinically relevant therapeutic modality, the double injection model has a critical drawback. A tumor sends out signals to affect the host (a review [35]), making it impossible to determine whether the change in lung environment is due to chemotherapy per se or chemotherapy plus signals from the primary tumor. The pre-treatment model does not suffer this drawback and is “cleaner.” To avoid confounding factors, we used the pre-treatment model in the rest of the manuscript to study mechanisms.

### 2.2. The Effect of CTX and the Host Atf3 on Cancer Cell Extravasation, Proliferation, and Death

For circulating cancer cells to form colonies in the lung, they need to extravasate (entering the tissue parenchyma) and grow (higher proliferation than cell death). To examine extravasation, we stained the lung sections by co-immunofluorescence for cancer cells (tGFP^+^), endothelial cells (ECs, CD31^+^), and nuclei (Topro). Figure 2a (right) shows an example of cancer cells either inside the blood vessel or outside (in the lung parenchyma). Images from all four groups of lungs were coded, pooled, reshuffled, and scored in a blind manner. As shown in Figure 2a (left), CTX did not affect extravasation in either WT or *Atf3* KO lungs. However, the host *Atf3* enhanced extravasation, as evidenced by the higher percentage of extravasated cells in WT than *Atf3* KO lung. Thus, the *Atf3* KO lung is less conducive for cancer cells to extravasate.

To investigate colony growth, we examined cancer cell proliferation and death. For proliferation, we co-stained the lung sections for tGFP (cancer cells) and phospho-histone H3, a marker for mitotic cells [36]. In general, neither CTX nor the host *Atf3* made any difference. That is, all four groups of lungs had similar percentage of proliferating cancer cells, with an exception at 9 h in KO lung where CTX decreased cell proliferation (Appendix A). This exception can contribute to a lower cancer cell burden in this group. For cancer cell death, we stained for tGFP (cancer cells) and activated caspase 3 (apoptosis). Figure 2b shows that *Atf3* KO lung had higher percentage of apoptotic cancer cells than WT lung at all time points examined, indicating that the KO lung is less hospitable than the WT counterpart. CTX treatment, in general, increased apoptosis in the KO lung, but not in the WT lung.

In summary, the *Atf3* KO lung has a less conducive environment for cancer cells to extravasate or survive, leading to an overall lower lung colonization in the *Atf3* KO than WT lungs. Initially, CTX enhanced cancer cell retention in both WT and *Atf3* KO lungs. However, this advantage in the KO lung dwindled away over time due to CTX-increased apoptosis. In contrast, WT lung treated with CTX did not suffer this dwindling effect, allowing the CTX-increased cancer cell retention to manifest. The net result is an increase in lung colonization by CTX in a host-*Atf3*-dependent manner.

### 2.3. Macrophage Is a Key Cell Type for the Host Atf3 to Mediate the Pro-Colonization Effect of CTX

To investigate the key host cell type for ATF3 action, we used conditional KO (CKO) mice deficient in *Atf3* selectively in macrophage (*Atf3*^f/f^, *LysM*-*Cre*, characterized in ref. [33]). The rationale for investigating *Atf3* in macrophage is two-fold: (a) macrophage plays an important role in cancer progression (reviews [4,37,38,39]); (b) macrophage is a key cell type for *Atf3* function [33,40]. Thus, it is possible that *Atf3* in macrophages plays an important role to mediate the pro-colonization effect of CTX. Figure 3a shows that the ability of CTX to enhance lung colonization was reduced in the *Atf3* CKO lung (*p* < 0.01). However, the reduction (by ~40%) was not as robust as that in the total *Atf3* KO mice (by ~95%), indicating that *Atf3* in other host cells also plays a role to mediate the pro-colonization effect of CTX. As controls, we examined *Atf3* induction in macrophages by CTX. In vitro, *Atf3* was induced in bone-marrow-derived macrophages (BMDMs) by 4-hydroperoxy cyclophosphamide (4-OOH), an active form of CTX (Appendix A). In vivo, the *Atf3* mRNA levels trended higher in monocytes/macrophages (CD11b^+^, Ly6G^−^) from CTX-treated lungs than that from PBS-treated lungs at 72 h after cancer injection (Appendix A). For earlier time points, we analyzed the lungs by co-immunofluorescence for ATF3 and F4/80 (a macrophage marker). As shown in Appendix A, the percentage of macrophages positive for ATF3 was higher in CTX-treated than PBS-treated lungs. Furthermore, the *Atf3* mRNA level trended higher in the circulating myeloid cells (CD11b^+^) at 16 h after CTX treatment (Appendix A). Note that this is before cancer cell injection. Thus, monocyte (a myeloid-lineage of cell and the macrophage precursor) has not been recruited to the lung yet, necessitating the analysis of blood (rather than lung) samples. All these results support the notion that CTX induces *Atf3* in monocytes/macrophages. Importantly, CTX induced *Atf3*—in the absence of cancer signals. Taken together, *Atf3* is induced during the pre-treatment and colonization periods and plays an important role to mediate the CTX effect. 

As a complementary approach to CKO, we carried out an add-back experiment by co-injecting cancer cells into the KO mice (the recipients) with myeloid cells (CD11b^+^) from four groups of lungs: WT or KO with CTX or control pre-treatment (diagrammed in Figure 3b). Figure 3c shows that WT myeloid cells mediated the effect of CTX to increase colonization, but the KO myeloid did not. In the recipient mice, all host cells—other than the injected myeloid cells—are *Atf3*^−/−^. Thus, *Atf3* in myeloid cells—without help from *Atf3* in other host cells—is sufficient to mediate some CTX effect. Together, data from CKO and add-back experiments indicate that *Atf3* in macrophage/myeloid is, at least partially, necessary and sufficient for CTX to exert its pro-colonization effect.

### 2.4. Cancer Cell Transendothelial Migration In Vitro Is Affected by the Genotype of Atf3 in Macrophages, but Not by CTX

As shown above, the host *Atf3* enhanced extravasation (Figure 2a). To test the role of *Atf3* in macrophage for this process, we established a transendothelial migration assay using Boyden chambers. We co-cultured tGFP-MVT1 cancer cells with bone-marrow-derived macrophages from WT or *Atf3* KO mice and assayed the ability of cancer cells to migrate through a monolayer of ECs as schematized in Figure 4a. The ECs were purified from mouse lungs by fluorescence activated cell sorting (FACS) to >90% purity (Appendix A) and are referred to as the mouse lung ECs (mLECs). As shown in Figure 4b, cancer cells co-cultured with the WT macrophages migrated through ECs more efficiently than that with KO macrophages (by 35–40%). Interestingly, the genotype of mLECS did not affect cancer cell migration (Appendix A). Therefore, *Atf3* in macrophages, but not ECs, affected cancer cell transendothelial migration. We then tested whether CTX affects transendothelial migration. Because CTX is a pro-drug (activated by liver metabolism), we injected mice with CTX, collected their sera, and applied the sera to the mLECs and macrophage co-culture, followed by washing and addition of cancer cells (detailed in Methods). Sera from PBS injected mice were used as control. As shown in Figure 4c, CTX did not affect cancer cell transendothelial migration, consistent with the in vivo data that CTX did not affect extravasation (Figure 2a).

### 2.5. Atf3 Dictates a Dichotomous Role of Macrophage in a Cell-Autonomous Manner 

To further investigate the role of *Atf3* in macrophages, we depleted macrophages in WT or KO mice by clodronate-encapsulated liposome [41], with liposome only as control (Figure 5a). Analysis of lung immune cells by flow cytometry followed by t-distributed stochastic neighbor embedding (tSNE), a dimensional reduction algorithm that groups cells with similar features into clusters, showed efficient depletion of macrophages (Figure 5b), which are characterized by CD11b^+^, F4/80^+^, CXCR3^+^, Ly6G^−^, Ly6C^−/lo^, Siglec-F^−^ (Appendix A). Figure 5c shows that depletion of macrophages in CTX-treated WT mice reduced lung colonization, indicating that those WT macrophages enhanced colonization. This result is consistent with previous reports that macrophages characterized by CD11b^+^, F4/80^+^, and CX3CR1^+^ facilitate pre-metastatic niche formation in the lung [31,42]. Intriguingly, depletion of macrophages in CTX-treated KO mice had an opposite effect, indicating that those KO macrophages repressed colonization. That is, the KO macrophages do not simply lack positive factors in the WT counterparts to enhance colonization; rather, they had negative factors to repress colonization. Thus, *Atf3* genotype dictates a dichotomous role of macrophages: upon CTX treatment, the *Atf3*^−/−^ macrophage is an overall negative (Yin) contributor, and *Atf3*^+/+^ a positive (Yang) contributor to lung colonization. In Section 2.6 and Section 2.7 below, we will present data suggesting that the negative role of *Atf3*^−/−^ macrophage (upon CTX treatment) is, at least in part, due to its lack of a dampener (*Atf3*) to repress cytotoxic genes that have cancer-killing potential (more in Discussion).

We also depleted macrophages in the CKO mice and found a similar result: depleting the *Atf3*^+/+^ macrophages (in the flox/flox control mice) decreased colonization, but depleting the *Atf3*^−/−^ macrophages (in the CKO mice) increased colonization (Figure 5d). In the above experiment using *Atf3* whole body KO mice, all host cells are *Atf3*^−/−^. In CKO mice, all host cells other than macrophages are *Atf3*^+/+^. Therefore, the ability of *Atf3* status in macrophages to dictate a dichotomous function is a cell autonomous feature, not affected by the genotype of *Atf3* in other host cells (to any appreciable degree). This cell autonomy, combined with ATF3 as a transcription factor, prompted us to investigate the transcriptome of macrophages.

### 2.6. The Effect of Atf3 Genotype and CTX on Macrophage Gene Expression

We sorted CD11b^+^, Ly6G^−^, cells from the lungs to enrich macrophages (>90% purity, Appendix A) from four groups of mice (WT or KO mice with CTX or control pre-treatment) and isolated their RNAs for gene expression profiling on a Clariom S array (Affymetrix Gene-Level Array). RNAs from eight to ten mice per group were combined to compensate for biological variations, and the data were analyzed by the Transcriptome Analysis Console software. We looked for genes that are regulated by CTX differentially in WT versus *Atf3* KO macrophages. A group of ~850 genes became evident; they are upregulated by CTX but with much higher induction in the KO than WT macrophages (more than twofold in KO cells), indicating that ATF3 dampens their induction. Gene ontology analysis indicated that this gene set is enriched in biological functions of defense against other organisms and negative regulation of growth (Figure 6a). Among them are genes involved in proinflammatory response fighting infection (Figure 6b). Of note are the antimicrobial genes cathelicidin (*Camp*), lipocalin 2 (*Lcn2*), and lactotransferrin (*Ltf*). They encode antimicrobial proteins—one of the top functional group of genes identified in the array (Figure 6a). In addition, they are among the genes with the highest differential regulation by CTX (more in the *Atf3* KO than WT macrophages). RT-qPCR confirmed their differential induction (Figure 6d). The gene ontology result suggests that group 4 macrophages (that is, KO macrophages from CTX treated mice) may have higher cytotoxicity than other groups, due to their higher expression of defense genes. In vitro analysis of cancer cell killing by four groups of lung macrophages confirmed this idea (Figure 7a,b). To test this in vivo, we carried out clodronate depletion coupled with apoptosis assay. As shown in Figure 7c, depleting macrophages from CTX-treated *Atf3* KO mice (group 4) resulted in less cancer cell apoptosis, indicating that these macrophages (*Atf3*^−/−^) are pro-apoptotic, consistent with their cytotoxicity in vitro. Interestingly, depleting macrophages from CTX-treated WT mice had an opposite effect, indicating that these macrophages are anti-apoptotic. Analysis of the gene expression data revealed that several M2-skewing genes are preferentially upregulated by CTX in the WT macrophages (Figure 6c). Note that many of them are novel (rather than the classic) M2 markers, such as *Cdh1* [43], *Irf4* [44], *Egr2*, and *c-Myc* [45]. Since M2-skewed macrophages are immune suppressive (inhibiting cytotoxic immune cells from killing cancer cells), this pattern is consistent with the nature of these macrophages (*Atf3*^+/+^ from CTX-treated mice, group 2) as anti-apoptosis for cancer cells. However, despite being anti-apoptotic in vivo, these macrophages were cytotoxic to cancer cells in vitro (Figure 7a). This apparent discrepancy can be explained by the differences between in vitro and in vivo conditions, such as the lack of cytotoxic immune cells in vitro.

### 2.7. The Antimicrobial Gene Cathelicidin in Cancer Cell Death

The three antimicrobial genes identified in the transcriptome analysis were induced by CTX, but the induction was much lower in WT than *Atf3* KO macrophages. This means that ATF3 dampened their induction, either directly or indirectly. The protein products of these genes are secreted molecules and inhibit microbes via different mechanisms. LCN2 and LTF are antimicrobial by sequestering iron (an essential nutrient for microbes) and/or modulating immune functions (reviews, [46,47]). CAMP has been shown to be cytotoxic to cells in close proximity to macrophages [48]. This direct cytotoxicity prompted us to test whether CAMP is functional in killing the MVT-1 breast cancer cells. As shown in Figure 6e, the active form of Camp gene product (LL-37) significantly increased cancer cell death. We also carried out co-immunofluorescence analysis of CAMP and F4/80. Appendix A shows a representative image. Quantification of the data showed that group 4 lung (CTX-treated *Atf3* KO mice) had the highest percent of macrophages positive for CAMP (Figure 6f), consistent with its highest cancer cell death (shown in Figure 2b). To test the relevance of Camp to human breast cancer, we analyzed publicly available gene expression data derived from breast cancer patients and examined the ratio between *Camp* and *Atf3* mRNAs (*Camp*:*Atf3*), where a high ratio (high *Camp*, low *Atf3*) imitates the KO scenario. As shown in Figure 8, the high *Camp*:*Atf3* ratio correlated with high patient survival. The same is true for the *Lcn2*:*Atf3* and *Ltf*:*Atf3* ratios. Together, they are consistent with the data that KO mice have lower cancer burden, lending credence to our mouse models.

## 3. Discussion

We propose the following model to explain how CTX increases lung colonization in the WT but not *Atf3* KO lung. In the WT lung, CTX increased the ability of lung to retain cancer cells upon their arrival. Although CTX made no difference in the lung environment to affect cancer cell extravasation, CTX-treated *Atf3*^+/+^ macrophages are anti-apoptotic, contributing to decreased cancer cell death. The overall consequence is the increased lung colonization by CTX in the WT lung. In the *Atf3* KO lung, CTX also increased the ability of lung to retain cancer cells upon their arrival. However, the *Atf3* KO lung is less conducive for cancer cells to extravasate or survive, leading to an overall lower lung colonization in the *Atf3* KO than WT lungs. Interestingly, the initial advantage of increased cancer cell retention by CTX dwindled away over time due to higher cancer cell apoptosis, a feature accompanied by the upregulation of defense genes and an increased cytotoxicity of macrophages toward cancer cells. The net result is no increase in lung colonization by CTX in the *Atf3* KO lung. In summary, the status of *Atf3* dictates a dichotomous (yin–yang) macrophage activity (upon CTX treatment): anti-apoptotic in the WT but pro-apoptotic in the *Atf3* KO macrophages. Although the dichotomous role of macrophages in cancer (promote or suppress it) is well documented (reviews, [49,50,51]), the new information here is threefold: (i) the status of a stress-inducible master-switch gene *Atf3* plays a pivotal role; (ii) the dichotomy is manifested under the treatment of cyclophosphamide, a frontline chemotherapeutic agent; (iii) the antimicrobial genes are likely a part of the transcriptional program associated with anti-cancer activities. We note that CTX does not affect cancer cell extravasation but the genotype of *Atf3* in macrophages does. Macrophages are known to affect the ability of cancer cells to migrate through endothelial cells, either by secreting soluble factors (a review, [52]) or by direct contact (examples, [53,54]). Since the transendothelial migration assay in this study had BMDMs in the bottom chambers, the effect we observed is via soluble factors. The genotype difference provides a handle for future mechanistic studies by comparing WT and *Atf3* KO macrophages.

In addition to macrophage activities, CTX also affected macrophage abundance. As shown in Appendix A, CTX increased macrophages (CD11b^+^, LyG6^−^, F4/80^+^) in WT, *Atf3* KO, and CKO lungs (at 72 h after cancer cell injection). This increase in abundance would augment the effect of CTX on macrophage properties (discussed above), accentuating the phenotype: lower cancer cell death in WT lung but higher in the KO counterpart. The effect of CTX on macrophage abundance can be due to several scenarios: (i) increase in progenitor cells in bone marrow, (ii) increase in recruitment and/or trafficking of the progenitors to the lung, (iii) increase in maturation, and (iv) any combination of them. Our preliminary results support scenario (i). As shown in Appendix A, CTX increased the common myeloid progenitor cells (CMPs, Lin^−^, Sca-1^−^, Kit-1^+^, IL7R^−^, CD34^+^, FcγR^−^, which give rise to all myeloid-lineage of cells) and granulocyte-monocyte progenitors (GMPs, Lin^−^, Sca-1^−^, Kit-1^+^, IL7R^−^, CD34^+^, FcγR^+^, which differentiate into monocytes, macrophages, and granulocytes) in bone marrow. Currently, it is not clear whether other scenarios are at work. More investigation is required to delineate the mechanisms.

Although the chemotherapeutic agents can affect both cancer cells and non-cancer host cells, the pre-treatment model as designed allowed us to determine how chemotherapy may affect the non-cancer host cells, which in turn affect the ability of cancer cells to colonize the lung. Our data showed that chemotherapy, by itself, without any signals from a primary tumor is sufficient to make the lung a cancer-friendly environment in a host-*Atf3*-dependent manner. Because *Atf3* is a stress-inducible gene and can be induced by various stress signals, including incision injuries (reviews, [24,27])—not just chemotherapeutic agents—our data have significant implications. It has been documented that surgery can enhance metastasis, a phenomenon dubbed as “therapy at a cost” [55]. Further, a case report documented a rapidly growing tumor at the site of minor head injury in a patient who had lung cancer treatment two years prior but with no signs of metastasis [56]. Mechanistic studies using mouse models demonstrated that injuries promote cancer growth by inducing extracellular matrix remodeling [57] and systemic inflammatory response [58]. Thus, changes in the tissue/host environment can re-activate dormant or clinically undetectable cancer cells to grow (a review, [59]). Since injuries and surgeries are stressors, we posit that these stressors induce *Atf3*, which in turn creates a cancer-friendly host environment (as shown in our data above), allowing residual cancer cells to thrive. In support of this supposition, *Atf3* is induced by various incision injuries (in skin, nerve, and liver; a review [27]). Since stress is unavoidable in life, a potential strategy to dampen stress-associated cancer progression is to counteract the consequences of *Atf3* induction (see below).

A potentially translatable finding in the report is the significantly higher induction of antimicrobial genes by CTX in the *Atf3* KO than WT macrophages. This finding indicates that CTX (a stressor) induces the antimicrobial genes and that this induction is dampened by ATF3. Since *Atf3* is also induced by CTX, ATF3 protein is a built-in dampener to prevent super-induction of antimicrobial genes. This is reminiscent of the role of *Atf3* in septic shock. *Atf3* is induced by septic shock. Upon induction, its gene product (ATF3 protein) reduces the expression of cytokine genes, which are also induced by septic shock. By doing so, ATF3 dampens cytokine storm and keeps stress response in check [40]. Several points are worth noting here. First, we did not observe the same set of cytokine genes shown in the septic shock studies, presumably due to the different stressors in the paradigms: CTX versus LPS. Second, the results of antimicrobial genes were unexpected and, thus, novel. They provided new insights to the roles of CTX and *Atf3* in macrophage biology. Third, although *Atf3* provides a protection against septic shock as evidenced by the lower rate of death in WT than *Atf3* KO mice [40], *Atf3* is not beneficial in the context of cancer and chemotherapy. ATF3 makes the WT macrophages more cancer friendly by dampening their cytotoxicity. Therefore, whether *Atf3* induction is beneficial or detrimental is context dependent. Fourth, *Atf3* gene product, as a transcription factor, modulates the expression of many genes. Therefore, it is probably not appropriate to interrupt the entire function of ATF3. Instead, modulating selective downstream events of ATF3 may be more beneficial. Our finding that the antimicrobial CAMP peptide kills cancer cells is consistent with previous reports [48,60] and suggests that adding CAMP is a potential way to counteract the pro-cancer effect of ATF3, thus improving chemotherapy.

## 4. Materials and Methods

### 4.1. Animal Studies

All mouse studies were approved by the Ohio State University Institutional Animal Care and Use Committee (protocol number 2008A0169, renewed April 30, 2020). Age- and gender-matched (6–10 week) FVB/N or C57BL/6 WT, *Atf3* KO, *Atf3*^f/f^, and *LysM-Cre*/*Atf3*^f/f^ mice were used for all experiments. WT mice (originally purchased from Taconic Biosciences) and genetically modified mice (*Atf3*^−/−^, *Atf3^f^*^/f^, *LysM-Cre*/*Atf3*^f/f^ described previously [33,61]) were maintained as separate colonies. To avoid potential variations due to genetic drift, every 12–18 months, genetically modified mice were backcrossed with their respective WT mice to obtain heterozygotes for two generations before making the appropriate homozygotes. Mice were monitored daily and excluded if they showed any overt signs of stress, illness, or fighting. Euthanasia was performed if mice displayed labored breathing, inability to ambulate, or deteriorating body conditions. Mice were euthanized via CO_2_ asphyxiation and cervical dislocation. The rate of mortality for experimental mice was <1 %, with no obvious pattern of adverse response to treatment or cancer cell injection. All mice were maintained in climate-controlled, ventilated, and barrier-housing facilities at the Ohio State University, accredited by the American Association for Accreditation for Laboratory Animal Care (AAALAC). Injections and sample collections were performed in alternate orders (WT-control, WT-treated, KO-control, KO-treated, and then reverse the order) in each experiment. Each control and experimental group was housed in their own cages, and all cages under experiment were located on the same rack. JDM, JF, and SS were aware of the group allocation during the experiments. For the pre-treatment lung colonization model, mice were intraperitoneally (i.p.) injected with either vehicle (2% DMSO in PBS) or cyclophosphamide (Cayman Pharmaceuticals) at 150 mg/kg body weight, or paclitaxel (PTX) (Millipore Sigma, Darmstadt, Germany) at 20 mg/kg or vehicle (cremophor EL: ethanol: PBS, 1:1:2). Injections of chemotherapeutic drugs were performed in the afternoon between 3 and 5 PM. Four days later, turbo green fluorescent protein (tGFP)-labeled MVT-1 (tGFP-MVT1) breast cancer cells (10^6^ cells in 100 µL PBS) or tGFP-labeled Met-1 (tGFP-Met1) breast cancer cells (2 × 10^6^ cells in 100 µL PBS) were intravenously (i.v.) injected into the tail vein. Lungs were collected at the indicated time points for analyses. For the neo-adjuvant mouse model, mice were anesthetized using vaporized isofluorane (5% for induction and 2% for maintenance), and then, 10^5^ mCherry-labeled MVT-1 cells were injected into the 4th mammary fat pad of female mice in a 20 µL mixture of 1:1 ice cold matrigel and Dulbecco’s modified Eagle’s medium (DMEM). Tumors were allowed to grow until day 7, when they were palpable. Mice were treated with CTX or PTX, followed by i.v. injection of tGFP-MVT1 cells four days later, and the lung was collected 72 h after cancer cell injection. The differentially labeled MVT-1 cells allowed us to distinguish between cancer cells derived from the tumor and those from i.v. injection.

### 4.2. Cell Culture and Treatment

tGFP-MVT1 and tGFP-Met1 cells described previously [33] were cultured in Dulbecco’s modified Eagle’s medium (DMEM) (Gibco) supplemented with 10% fetal bovine serum (FBS), 1% penicillin/streptomycin (P/S) (Gibco), and 3.5 µg/mL puromycin (Sigma). These cell lines express two transgenes (tGFP and puromycin resistance genes), separated by internal ribosome entry site. For cyclophosphamide treatment in vitro, cells were cultured in a 12-well plate in 1 mL of growth medium containing 10 µM of 4-hydroperoxy cyclophosphamide (4-OOH) (Cayman Chemical) for the indicated time before analysis. For CAMP treatment, cells in a 96-well plate were cultured in 100 µL of growth medium containing 1–50 µM of CAMP peptide (Hycult Biotech), as indicated for 24 h before analysis.

### 4.3. Cell Isolation from Mouse Tissues

Mouse lung endothelial cells (mLECs) were isolated via a procedure adapted from two protocols [62,63]. Briefly, lungs from 7–8-day-old FVB/N neonatal mice were aseptically harvested and placed into Miltenyi C-tubes (Miltenyi Biotec) with 4 mL of a collagenase (Sigma) and dispase (Sigma) solution at 1 mg/mL each and then minced into 2 mm^3^ pieces. This mixture was incubated at 37 °C with constant agitation in a slowly oscillating water bath for 30 min, whereupon DNase I (Sigma) was added to a final concentration of 25 units/mL, followed by 30 min more of incubation. The tissue was then pulverized using a gentleMACS Dissociator (Miltenyi Biotec). The resulting slurry was passed through a 100 µm filter. Cells from three neonatal lungs were pooled and pelleted at 500× *g* at 4 °C and then resuspended in a 1× ammonium chloride red blood cell lysis buffer (15.5 mM NH4Cl, 1 mM KHCO3, 0.01 mM EDTA). After 10 min at room temperature, 10 volumes of water were added, and the cells were again pelleted at 500× *g* and resuspended in 90 µL of MACS buffer (0.5% bovine serum albumin and 2 mM EDTA in PBS) to generate single cell suspension (in general, ~3 × 10^6^ cells/neonatal lung) and mixed with 10 µL of CD31 magnetic beads, washed, and recovered in 1 mL of PBS as recommended (Miltenyi Biotec). About 5 × 10^5^ cells were recovered per neonatal lung, and ~1–3 × 10^6^ bead-enriched cells were cultured on 2% gelatin-coated 10 cm plates with EndoGRO LS medium containing 1% P/S for three days to reach confluence, at which time the purity was in general ~90% CD31^+^ by flow cytometry. mLECs were used for only three passages following isolation.

Lung macrophages (CD11b^+^, Ly6G^−^) were enriched from the adult mouse lungs by magnetic beads as follow. Single cell suspension from the adult lung was generated using gentleMACS Dissociator as described above (in general, ~10^7^ cells/adult lung). Approximately ~10^7^ cells in 90 µL of MACS buffer was mixed with 10 µL of Ly6G magnetic beads. The unbound fraction was incubated with 10 µL of CD11b magnetic beads, and the bound cells were recovered using magnetic columns as above.

Bone-marrow-derived macrophages (BMDMs) were isolated as described previously [33]. Briefly, femurs and tibia from the hind limbs of euthanized mice were removed, cleaned with 70% ethanol, and washed in PBS under sterile conditions. The ends of the bones were cut off, and the bones were flushed with PBS using a needle and syringe through a 100 µm filter. Cells were pelleted at 500× *g* at 4 °C and then resuspended in a 1X ammonium chloride red blood cell lysis buffer (described above). Bone marrow cells were cultured in differentiation medium (DMEM supplemented with 10% FBS, 1% P/S, and 20% conditioned medium from L929 cells). L929 conditioned medium was used as these cells secrete high levels of CSF-1, which promotes macrophage differentiation and survival. Bone marrow cells were cultured in this medium for two days, and the unattached cells were harvested, spun down at 500× *g* at 4 °C, and then plated in fresh differentiation medium on non-tissue culture dishes for seven days. Differentiated macrophages, appeared as elongated spindles, were used within 5–7 days before becoming a fried-egg shape.

### 4.4. Macrophage Cytotoxicity Assay

Lung macrophages (as described in cell isolation above) were plated onto 96-well plates pre-seeded with 25,000 tGFP-MVT1 cells at the ratio of 1:1 or 5:1 (macrophage: cancer). Cells were co-cultured for 18 h, and cancer cells were assayed for apoptosis.

### 4.5. In Vitro Apoptosis Assay

Cultured cells were washed and resuspended in 1× Binding Buffer (BD Biosciences), followed by staining with Annexin V at 500 ng/mL (ThermoFisher) and propidium iodide (PI) at 1 µg/mL (Sigma) for 5 min, followed by flow cytometry. For some experiments, tGFP-labeled cancer cells were used to distinguish cancer cells from other cells in the culture.

### 4.6. Image Analysis

For all image analyses, six images (at 100× magnification) were collected per lung section. Images from all groups of mice were coded, pooled, reshuffled, and analyzed in blind to prevent bias. Randomization was performed using the “Random Names” tool by Jason Faulkner. Individual colony was defined as a cluster of cells within 10 µm of each other. Analysis was done on Zeiss Zen Lite software or FIJI where appropriate.

### 4.7. RNA Isolation and RT-qPCR

RNA was isolated from the indicated cells using TRIzol (ThermoFisher) according the manufacturer’s instructions. Concentration and purity were determined by NanoDrop 2000 (ThermoFisher). RT-qPCR was performed as previously described [64] using GAPDH as an internal control. All primers are listed in Appendix A.

### 4.8. Generation of CTX-Serum and PBS-Serum

FVB/N mice were injected with PBS or CTX. Sixteen hours later, blood (500 µL per mouse) was collected by cardiac puncture and left at room temperature for 30 min to clot, followed by centrifugation at 500× *g* for 10 min at 4 °C. The resulting supernatant (~200 µL) was collected as serum and stored at −80 °C. They are referred to as PBS serum or CTX serum. The 16 h time point was chosen to allow time for CTX to be metabolized by the liver and for secreted molecules (if any) to accumulate in the serum in response to CTX.

### 4.9. Transendothelial Migration Assay

BMDMs and mLECs from WT and *Atf3* KO mice were isolated as described above. Multiple mice (3 to 5) from each group were used to compensate for biological variation. BioCoat Matrigel 12-well Invasion Chambers (Corning) were warmed to room temperature from −20 °C and rehydrated with 500 µL of PBS for two hours at 37 °C. Afterwards, 5 × 10^4^ mLECs (WT or *Atf3* KO) were plated on the top chamber with 500 µL of EndoGro LS medium supplemented with 2% FBS and cultured for two days to reach confluency. An amount of 10^5^ BMDMs were plated in the bottom chamber in 1 mL of DMEM medium supplemented with 10% FBS and 1% pen/strep. The next day, 10^5^ tGFP-MVT1 cells were plated on top of mLECs, and 24 h later the underside of the membrane separating the two chambers were imaged (Nikon Diaphot inverted microscope) to analyze the cancer cells that had migrated through the mLECs. Multiple wells were used for each group and all images (100× magnification, 5–9 fields per membrane) were coded, pooled, reshuffled, and counted in blind. For testing the effect of CTX, after mLECs and BMDMs had been cultured, PBS-serum or CTX serum (20 µL) generated above was applied to the top and bottom chambers for 24 h. After this, the medium was removed, wells were washed with PBS, and then incubated with fresh medium for one day before tGFP-MVT1 cells (10^5^) were plated on top of mLECs, followed by analysis of the membrane 24 h later.

### 4.10. Immunofluorescence Microscopy

Staining was performed on formalin-fixed, paraffin-embedded tissue sections (10 µm thickness) as previously described [65]. Briefly, slides were rehydrated, blocked with 0.3% hydrogen peroxide in methanol, washed in Tris-buffered saline + 0.1% Tween 20 (TBST), and blocked in 5% normal horse serum for an hour at room temperature. Primary antibodies were applied at the indicated concentrations (Appendix A) overnight at 4 °C. Slides were then incubated with ImmPress secondary antibodies (Vector) for an hour, followed by the application of a tyramide signal amplification system (PerkinElmer) for 15 min, both at room temperature. Slides were then mounted in Vectashield (Vector). For the determination of intravascular/extravascular cells, cancer cells were only deemed outside the blood vessel if a nucleus associated with tGFP staining was completely outside of CD31^+^ blood vessels. Images were captured on a Leica TCS SL confocal microscope, an Andor Spinning Disk confocal microscope, or a Zeiss LSM800 confocal super-resolution microscope (depending on availability).

### 4.11. Analytical Flow Cytometry and Fluorescence Activated Cell Sorting (FACS)

Single cell suspensions from lungs were prepared as described in Cell Isolation. Afterward, 2 × 10^6^ cells were incubated with CD16/32 Fc blocking antibodies for 10 min at 4 °C in 100 µL of flow buffer (5% FBS in PBS). Cells were then pelleted in a refrigerated centrifuge at 500× *g* for 5 min, resuspended in flow buffer, and stained with the indicated antibodies for 15 min at 4 °C. Antibodies were washed away, and the cells were fixed in 2% formalin (in PBS) for at least 30 min, pelleted, and resuspended in flow buffer before analysis. Cells were analyzed on BD LSR II or Fortessa flow cytometers, depending on availability. For FACS, cells were run on a BD FACSAria III or a BD FACSAria Fusion on a purity setting. Unstained and single-stained cells were used as gating controls and for compensation. T-distributed stochastic neighbor embedding (tSNE) was performed using the Flow Jo platform. Compensated samples were concatenated into one overall sample. The samples were run through 1000 iterations with a perplexity of 30 using a vantage point tree k-nearest neighbors algorithm and a Barnes–Hut gradient algorithm.

### 4.12. Macrophage Depletion Assay

Mice were pre-treated with CTX or vehicle and injected with cancer cells as described before, except with the addition of i.v. injections of clodronate-encapsulated liposomes or control liposome (200 µL) (Liposoma BV) at 18 h before and 2 h after cancer cell injection. The liposomes were injected in the opposite tail vein of the cancer cell injections. The effectiveness of depletion was determined by analyzing the lung cells using flow cytometry after staining for CD11b, Ly6G, Ly6C, F4/80, CX3CR1, NK1.1, and Siglec-F.

### 4.13. Microarray

CD11b^+^, Ly6G^−^ cells were sorted from mouse lungs and RNAs isolated using TRIzol, followed by analysis on the Clariom S mouse microarrays (Applied Biosystems) at the Genomics Shared Research core facility at OSU. Eight to ten mice per group were used. Data were analyzed using Transcriptome Analysis Console (Applied Biosystems) and genes demonstrating a ≥ 1.5 fold difference were included for further analysis. Gene ontology was performed using the online tool DAVID Bioinformatics Resource 6.8 (https://david.ncifcrf.gov/ (accessed on 8 August 2020)).

### 4.14. Public Dataset Analysis

Global gene expression datasets from The Cancer Genome Atlas (TCGA; https://cancergenome.nih.gov/ (accessed on 4 December 2020) were analyzed by PROGgeneV2 [66] to determine potential correlation between patient survival and the ratio of target gene to *Atf3* expression.

### 4.15. Statistics

The sample size was calculated based on the standard deviation in previously published results [19,33] or pilot experiments, with the expectation to detect a 20% difference (2-sided) with 80% power and 95% confidence. Cancer burden or colony number was used as the outcome measurement, depending on the experiment. Data were analyzed using GraphPad Prism 6.0 and SigmaPlot 13.0. GraphPad was used for the generation of figures, and SigmaPlot for statistics. All data represent mean ± standard error (SE). A *p* value of less than 0.05 was considered statistically significant. Student’s *t* test, one-way ANOVA, and two-way ANOVA were used as indicated. A post hoc Holm–Šídák correction to counteract family-wise error rate was used for all ANOVA tests, and normality was assessed using the Shapiro–Wilk test. Grubb’s test was used to detect outliers, and none were found for all data reported here.

## 5. Conclusions

Data in this report provide mechanistic explanations for the paradoxical ability of CTX to exacerbate lung colonization in a host-*Atf3*-dependent manner. One key mechanism is the ability of *Atf3* to dictate a dichotomous (yin-yang) switch of CTX-treated macrophage: from pro-cancer for WT to anti-cancer for *Atf3* KO macrophage. Since *Atf3* is a stress-inducible gene, our data suggest that how macrophage responds to stress plays an important role in the paradox of chemotherapy. Because the anti-cancer activity of the CTX-treated *Atf3* KO macrophage is associated with its high expression of cathelicidin, a cytotoxic antimicrobial peptide, we posit that cathelicidin may have the potential to dampen the pro-cancer effect of chemotherapy.

## Figures and Tables

**Figure 1 ijms-22-07356-f001:**
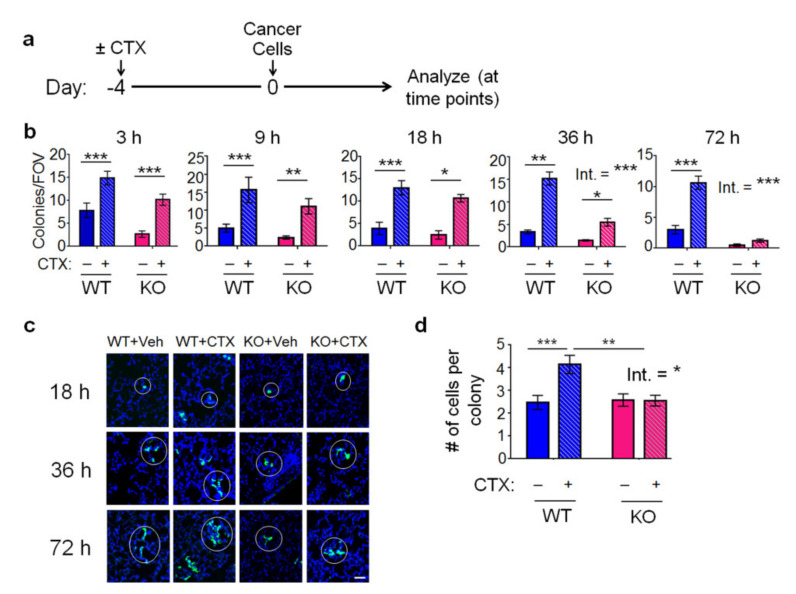
Host *Atf3*, combined with CTX, establishes a cancer-friendly tissue environment at early stages of lung colonization. (**a**) A schematic of the lung colonization model with CTX (+) or vehicle (−) treatment on day 4 before tail vein injection of the tGFP–MVT1 breast cancer cells to WT or *Atf3* KO mice. (**b**) Lungs were collected at the indicated time points after cell injection, stained by immunofluorescence for tGFP, and the colony number per field of view (FOV) scored at the indicated time points. *n* = 3–5 for 3 h (3 for vehicle groups, 5 for CTX groups); 6–11 for 9 h (6 for WT + Veh, 11 for KO + Veh, 7 for CTX groups); 7–13 for 18 h (8 for WT + Veh, 7 for WT + CTX, 11 for KO + Veh, 13 for KO + CTX); 7–13 for 36 h (12 for WT + Veh, 13 for WT + CTX, 8 for KO + Veh, 7 for KO + CTX); 9–10 for 72 h (9 for Veh groups, 10 for CTX groups); from 2–3 independent experiments. (**c**) Representative images of individual cancer colonies (circled in white) from four groups of mice: WT or KO mice with CTX or vehicle (Veh) treatment, referred to as WT + Veh, WT + CTX, KO + Veh, and KO + CTX. Green: tGFP (cancer cells); blue: Topro (nuclei). (**d**) Numbers of cancer cells per colony at 72 h after cell injection (*n* = 9–10, from 3 independent experiments). Scale bar, 20 µm. Bars indicate mean ± SEM; two-way ANOVA with post hoc Holm–Šídák correction; Int.: treatment–genotype interaction; Int.: treatment–genotype interaction; ** p* < 0.05; ** *p* < 0.01; *** *p* < 0.001.

**Figure 2 ijms-22-07356-f002:**
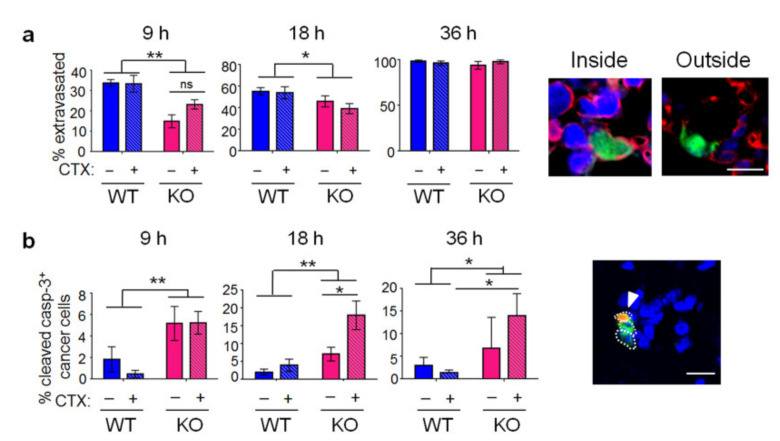
The effect of CTX and the host *Atf3* on cancer cell extravasation and apoptosis. (**a**) The lungs were co-stained by immunofluorescence for cancer cells (tGFP^+^, green), endothelial cells (CD31^+^, red), and Topro (nuclei, blue). Left: The percent (%) of cancer cells outside the blood vessel at the indicated time points. *n* =6–12 for 9 h (6 for Veh groups, 7 for WT + CTX, 12 for KO + CTX); 5–7 for 18 h (6 for WT + Veh, 6 for WT–CTX, 5 for KO + Veh, 7 for KO + CTX); 3–4 for 36 h (3 for WT + Veh, 3 for WT + CTX, 3 for KO + Veh, 4 for KO + CTX); from 2–3 independent experiments. By 36 h, ~100% of cancer cells have extravasated in all groups of lungs. Right: Representative images showing cancer cells inside or outside the blood vessel. (**b**) The lungs were co-stained by immunofluorescence for cancer cells (tGFP^+^, green), activated caspase 3 (red), and Topro (nuclei, blue). Left: The % of cancer cells positive for activated caspase 3. *n* = 6–9 for 9 h (7 for WT + Veh, 8 for WT + CTX, 9 for KO + Veh, 6 for KO + CTX); 6–13 for 18 h (7 for WT + Veh, 6 for WT + CTX, 6 for KO + Veh, 13 for KO + CTX); 4–9 for 36 h (7 for WT + Veh, 9 for WT + CTX, 4 for KO + Veh, 8 for KO + CTX); from 2–3 independent experiments. Right: Representative images. Dotted circles indicate three cancer cells: two negative for activated caspase 3 and one positive (arrowhead). Scale bar, 20 µm. Bars indicate mean ± SEM; two-way ANOVA with post hoc Holm–Šídák correction; ns: not significant, * *p* < 0.05; ** *p* < 0.01.

**Figure 3 ijms-22-07356-f003:**
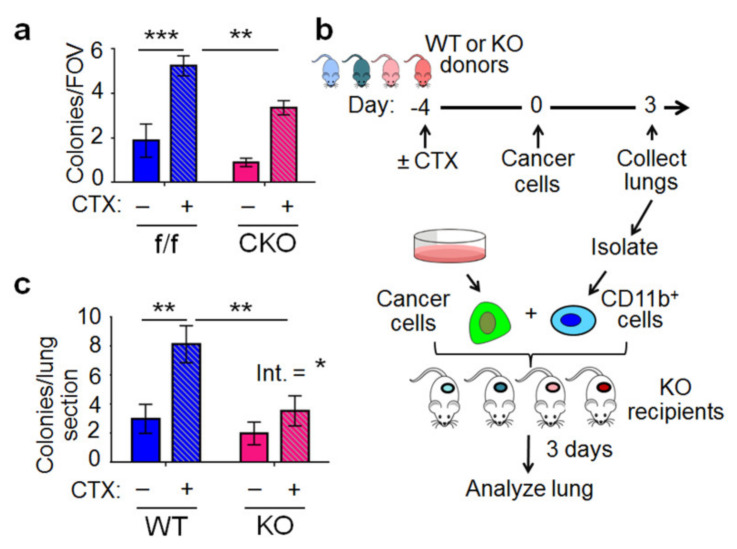
Macrophage is a key cell type for the host *Atf3* to mediate the pro-colonization effect of CTX. (**a**) *Atf3* conditional KO (CKO) and control flox mice (f/f) were treated with CTX (+) or vehicle (−) 4 days before cancer cell injection as schematized in Figure 1a, and the lung colony numbers were analyzed at 72 h. *n* = 8–28 (7 for WT + Veh, 23 for WT + CTX, 8 for KO + Veh, 28 for KO + CTX); from 5 independent experiments). (**b**) A schematic of the add-back experiment that co-injected into the *Atf3* KO recipient mice with cancer cells and myeloid cells (CD11b^+^) isolated from the lungs of four groups of donor mice: WT + Veh (light blue); WT + CTX (dark blue); KO + Veh (light red); KO + CTX (dark red). Lungs were collected and analyzed 3 days later. (**c**) Colony numbers per lung section in the recipient mice at 72 h after cell injection. *n* = 10–13 (10 for WT + Veh, 10 for WT + CTX, 8 for KO + Veh, 13 for KO + CTX); from 3 independent experiments. Bars indicate mean ± SEM; two-way ANOVA with post hoc Holm–Šídák correction; Int.: treatment–genotype interaction; * *p* < 0.05; ** *p* < 0.01; *** *p* < 0.001.

**Figure 4 ijms-22-07356-f004:**
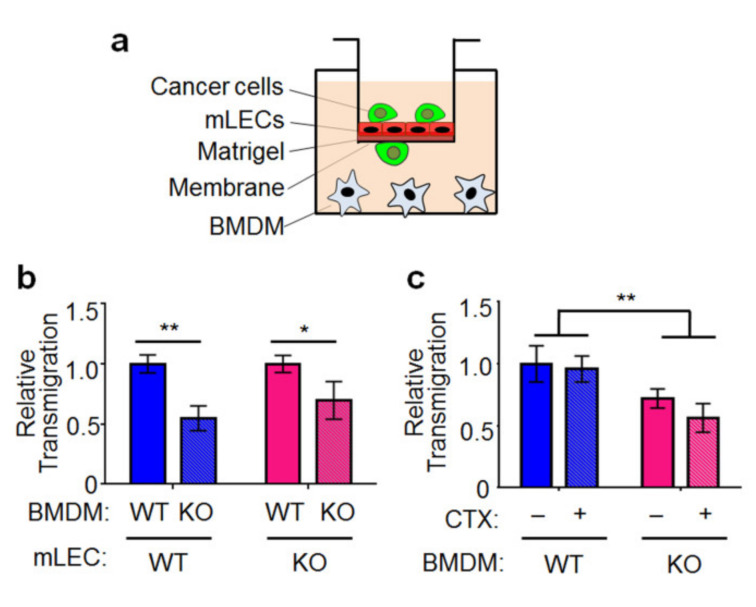
Cancer cell transendothelial migration in vitro is affected by the genotype of *Atf3* in macrophages, but not by CTX. (**a**) A schematic of the transendothelial migration assay. Cancer cells: tGFP-MVT1; BMDM: bone-marrow-derived macrophage from WT or *Atf3* KO mice; mLEC: mouse lung endothelial cell from WT or KO mice. (**b**) WT or *Atf3* KO BMDMs were co-cultured with a monolayer of WT or KO mLECs (as indicated), and the cancer cells on the underside of the membrane were counted 24 h later (see Methods for details). Cell number from the WT BMDM groups was arbitrarily defined as 1 (*n* = 5, from 2 independent experiments). (**c**) WT or *Atf3* KO BMDMs were co-cultured with a monolayer of WT mLEC, then treated with serum from CTX-treated mice (CTX+; which contains active CTX metabolites) or serum from vehicle-treated mice (CTX-), and the migrated cancer cells counted as in (**b**). The number from WT BMDM with control serum (CTX-) was arbitrarily defined as 1 (*n* = 6, from 2 independent experiments). Bars indicate mean ± SEM; two-way ANOVA with post hoc Holm–Šídák correction; * *p* < 0.05; ** *p* < 0.01.

**Figure 5 ijms-22-07356-f005:**
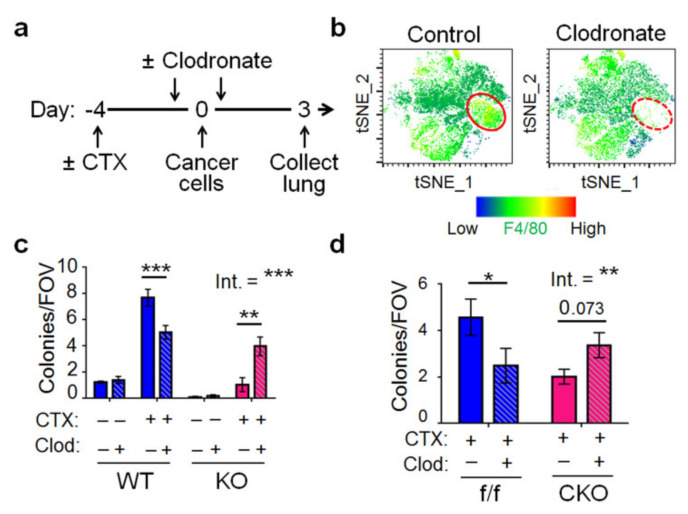
*Atf3* dictates a dichotomous role of macrophage in a cell-autonomous manner. (**a**) A schematic for the clodronate (Clod) depletion experiments. (**b**) A representative tSNE of flow cytometry analysis of lungs from mice treated with clodronate or control liposome, with the signal intensity for F4/80 shown. Red circles indicate the macrophage population of cells (CD11b^+^, Ly6G^lo^, Ly6C^lo^, F4/80^+^, CX3CR1^+^, Siglec-F^-^). See Appendix A for details on cell markers. (**c**) Four groups of mice, WT or *Atf3* KO mice with CTX (+) or vehicle (−) pre-treatment, were established; each group was then split into clodronate (+) or control liposome (−) treatment as schematized in (**a**). Lungs at 72 h after cancer cell injection were analyzed for colony number. *n* = 7–13 (9 for control liposome groups, 7 for WT + Veh + Clod, 10 for WT + CTX + Clod, 12 for KO + Veh + Clod, 11 for KO + CT + Clod); from 3 independent experiments. (**d**) Same as (**c**) except that *Atf3* CKO and control flox (f/f) mice were used and that all mice received CTX pre-treatment. *n* = 6–10 (6 for WT + CTX + Control, 8 for WT + CTX + Clod, 11 for KO + CTX + Conrol, 10 for KO + CTX + Clod); from 3 independent experiments. Bars indicate mean ± SEM; two-way ANOVA with post hoc Holm–Šídák correction; Int.: treatment–genotype interaction; * *p* < 0.05; ** *p* < 0.01; *** *p* < 0.001.

**Figure 6 ijms-22-07356-f006:**
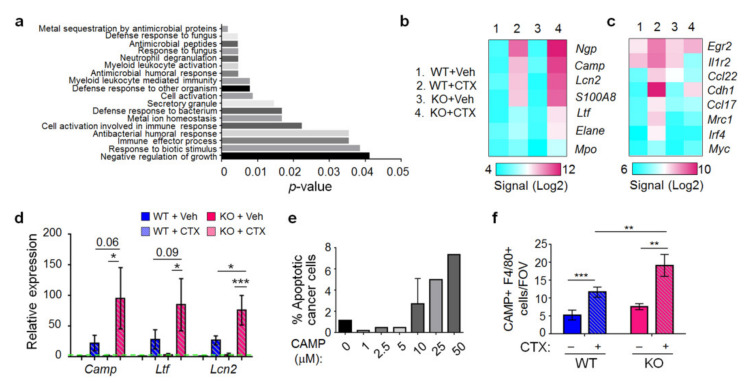
Gene expression profiling identified antimicrobial genes differentially upregulated by CTX in WT versus *Atf3* KO macrophages. Lung macrophages were enriched (CD11b^+^, Ly6G^−^) from four groups of mice (WT or *Atf3* KO, with CTX or vehicle pre-treatment) at 72 h after cancer cell injection, and their RNAs analyzed for gene expression on a Clariom S array (Affymetrix Gene-Level Array). (**a**) Transcriptome Analysis Console software identified a group of ~850 genes upregulated by CTX with higher induction in *Atf3* KO than WT macrophages. Shown is a subset of the enriched biological functions (based on gene ontology generated from the DAVID Functional Annotation Tool). (**b**) Heat map of the top genes differentially regulated by CTX in WT versus *Atf3* KO macrophages. *Mpo*: myeloperoxidase; *Elane*: elastase neutrophil expressed; *Ltf*: lactotransferrin; *Lcn2*: lipocalin 2; *Camp*: cathelicidin antimicrobial peptide; *Ngp*: neutrophilic granule protein. (**c**) Heat map of M2-skewing genes differentially regulated by CTX in WT versus *Atf3* KO macrophages. *Egr2*: Early growth response 2; *Il1r2*: interleukin 1 receptor type 2; *Ccl22*: C-C motif chemokine ligand 22; *Cdh1*: cadherin 1; *Ccl17*: C-C motif chemokine ligand 17; *Mrc1*: mannose receptor c-type 1; *Irf4*: interferon regulatory factor 4; *c-Myc*: cellular Myc proto-oncogene. (**d**) RNAs from the enriched macrophages (CD11b^+^, Ly6G^−^) were analyzed by RT-qPCR for *Camp*, *Ltf*, and *Lcn2* as indicated. Signals were standardized against that of *Gapdh*, and the average level in the WT + Veh group was arbitrarily defined as 1 (indicated by dotted line; *n* = 5 mice, from 5 independent experiments). (**e**) tGFP-MVT1 cancer cells were treated with cathelicidin peptide (LL-37, the mature form of Camp gene product) at the indicated concentration and analyzed 24 h later for apoptosis by Annexin V and propidium iodide (PI) stain. Percent (%) apoptosis is calculated by dividing the number of cancer cells in early (Annexin V+, PI−) and late (Annexin V+, PI+) apoptosis by total cancer cell number (*n* = 5–7, from 2–3 independent experiments). (**f**) Lung sections at 72 h after cancer cell injection were analyzed by co-immunofluorescence staining for cathelicidin (CAMP) and F4/80. Percent (%) of macrophages (F4/80^+^) positive for CAMP was scored in blind as detailed in Methods. *n* = 4–10 mice (4 for Veh groups, 10 for WT + CTX, 5 for KO + CTX); from 3 independent experiments. Bars indicate mean ± SEM; one-way (for panel e) or two-way (for panels d and f) ANOVA with post hoc Holm–Šídák correction; * *p* < 0.05; ** *p* < 0.01; *** *p* < 0.001.

**Figure 7 ijms-22-07356-f007:**
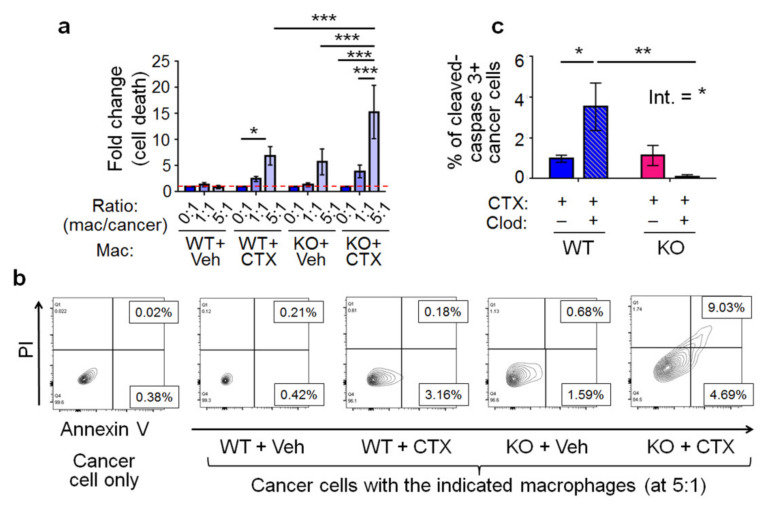
The cytotoxicity of macrophages in vitro and in vivo. (**a**) Lung macrophages (Mac, CD11b^+^, Ly6G^−^) were enriched from four groups of mice (WT or *Atf3* KO mice with CTX or vehicle pre-treatment) at 72 h after cancer cell injection and co-cultured with the tGFP-MVT1 cancer cells at the indicated ratio (mac/cancer) for 18 h. Cancer cells were analyzed by flow cytometry after Annexin V and PI staining, and the % of apoptosis is calculated as in Figure 6e. The numbers were then normalized by arbitrarily defining the control sample (cancer cells only, 0:1 ratio) as 1 (indicated by dotted line; *n* = 6, from 3 independent experiments). (**b**) A representative flow cytometry graph for different groups is shown. (**c**) WT and *Atf3* KO mice with CTX pre-treatment were treated with clodronate (+) or control liposome (-) and injected with cancer cells as schematized in Figure 5a. Lung sections at 72 h after cancer cell injection were analyzed by co-immunofluorescence and shown is the % of cancer cells positive for activated caspase-3. *n* = 5–8 mice (8 for Veh groups, 7 for WT + CTX + clodronate, 5 for KO + CTX + clodronate); from 2 independent experiments. Bars indicate mean ± SEM; two-way ANOVA with post hoc Holm–Šídák correction; Int.: treatment–genotype interaction; * *p* < 0.05; ** *p* <0.01; *** *p* < 0.001.

**Figure 8 ijms-22-07356-f008:**
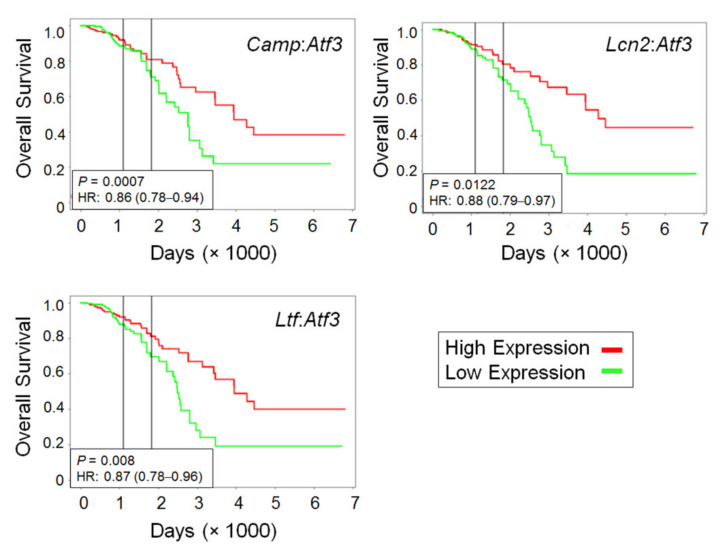
Analysis of gene expression data from human breast cancer samples. Publicly available microarray datasets (TCGA) were analyzed for the ratio between antimicrobial genes (*Camp*, *Ltf*, or *Lcn2*) and *Atf3*. Numbers above median were arbitrarily defined as high (red trace) and below median as low (green trace). The online tool PROGene V2 was used to generate the Kaplan–Meier curves of overall survival, which indicate that a high ratio (high antimicrobial but low *Atf3* gene expression) correlates with better outcome. Log-rank test; P values and hazard ratios (HR) are indicated.

## Data Availability

Microarray data have been deposited in the NCBI Gene Expression Omnibus (GEO) repository under accession number GSE164611 (https://www.ncbi.nlm.nih.gov/geo/query/acc.cgi?acc=GSE164611 (accessed on 14 January 2021)).

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
