# Peer review of "Stress-Inducible Gene *Atf3* Dictates a Dichotomous Macrophage Activity in Chemotherapy-Enhanced Lung Colonization"

_ijms, 2021, doi:10.3390/ijms22147356_

Round 1
Reviewer 1 Report
In their manuscript entitled “Stress-inducible gene Atf3 dictates a dichotomous macrophage activity in chemotherapy-enhanced lung colonization”, Middleton et al. investigate an under-appreciated contribution of chemotherapy to distal colonisation of cancer cells. The authors had previously reported that this is mediated in a manner that is dependent to host Atf3, a stress inducible gene. In the current study, the authors investigated in detail the dynamics of the increased colonisation and observed that treatment with the chemotherapeutic cyclophosphamide (CTX) increased the retention of circulating cancer cells in lung vascular bed in their model. Using conditional Atf3 knockout mice, they showed that although the absence of Atf3 had no impact on the CTX-induced increase in cancer retention or their extravasation into lung, the surrounding macrophages switches from a pro-tumorigenic to an anti-tumorigenic phenotype. This resulted in no net gain in lung colonisation, whereas the pro-tumorigenic macrophages in wild type mice would translate the increase in cancer cell retention into increase lung colonisation. The authors concluded that Atf3 plays a dichotomous role in promoting the formation/recruitment of pro-tumorigenic macrophage while suppressing cytotoxic macrophages.
This is an in-depth study that involves complicate experimental models and approaches. However, the work appears to be rigorously conducted with well designed controls. One of the major strengths of this manuscript is the way it is written — intuitively with simple English while avoiding being bogged down in spelling out everything. This makes for much easier reading and would help take the readers through the complexity of the data. Its biggest drawback is that the conclusion drawn are probably less exciting than the authors had hoped, though due to no fault of their own. Despite this, the authors did an admirable job in clearly describing the rationale of each experiment and its outcome; documenting their data painstakingly. However, all things considered, this is a difficult paper to understand, primarily because the experimental approaches are so complex, with a lot of attention to detail but often less than clear cut outcome. Furthermore, although the authors tried hard but the candidate genes they have highlighted in their gene expression profiling experiment (Fig. 6) do not fully explain the increased cytotoxicity of Atf3 KO macrophages.
Although I find the manuscript complete within its scope and having sufficient data to support its main conclusion, the authors are advised to address the following comments in a revised manuscript:
Fig. 1 - Have the authors any data that show that the length of the delay post CTX treatment affects lung colonisation?Fig. 1b - Was the rapid depletion of colonies in KO mice due to extravasation or cytotoxic activities of KO macrophages?Fig. S2 - The rationale of the “double injection” experiment is not clear. What was the original reason for the authors’ establishment of a “primary tumour” in the mammary fat pad? Did the experiment just not work as intended?
Fig. 2a - It may be confusing to the readers when this figure shows that cells have all extravasated within 36h while there are still colonies after 72h in Fig.1b;
Fig. 3b - Have the authors also tried to “add-back” KO donors-derived macrophages into WT recipients? Is the increased cytotoxic activity of the KO macrophages also cell-intrinsic, not needing the absence of Atf3 in the host cells?
Section 2.5 - I feel the authors have interpreted their data too rigidly at this point. Since “loss of Atf3” is not a natural state, it does not make sense to say “the loss of Atf3 have conferred increase cytotoxic activity to Atf3 KO macrophages” per se. It is better to reason that Atf3 acts as a negative regulator of immune surveillance; or indeed act to dampen the cytotoxic response as argued in the Discussion.
Fig. 6 - The authors tried hard to elucidate the effectors of the increased cytotoxicity of Atf3 KO macrophages, but the candidate genes they highlighted are unexpected. Moreover, the absence of differential cytokine expression between the 4 subsets is inconsistent with the notion that Atf3 acts to “dampen cytokine storm” during sepsis. Is it possible that some of the samples have become exposed to bacterial contaminants or LPS and hence the induction of anti-microbial genes in Atf3 KO macrophages? Can the authors provide data that exclude this possibility?
Lastly, the authors should address the possibility that the loss of Atf3 alters the differentiation or polarisation of macrophages post-CTX treatment, using well established markers.
Minor comments:Although the manuscript is well written, there are several places where the authors have overly simplified or abstracted the description. Some of these are listed below and should be corrected:
- p.3 line 118-120: “We also tested Met1 breast cancer cells, which differ from MVT-1 in their oncogenic events: Met1 derived from polyoma middle T antigen (PyMT), whereas MVT-1 from Myc and human VEGF.” - This is a problematic statement and should be edited. For example, Met1 is not a cell line derived from the PyMT, but rather from a tumour wherein the PyMT is the driver;
- p.3 line 124: “…injected with PyMT breast cancer cells derived from C57BL/6” - similar problematic language used here.
Typographical errors:- p.3 line 104: “… CTX increased lung colonization IN a host-Atf3 dependent manner”
Author Response
Point-by-point response for manuscript #ijms-1216872 (Middleton et al.)
REVIEWER #1:
Critique 1: Fig. 1 -… any data that the length of the delay post CTX treatment affects lung colonisation? Fig. 1b - Was the rapid depletion of colonies in KO mice due to extravasation or cytotoxic activities of KO macrophages? Fig. S2 - The rationale of the “double injection” experiment is not clear. What was the original reason …?
Response:
(a) Fig. 1: We have data on injecting cancer cells at 2 days post-CTX treatment. The results are similar but not as robust as those from 4 days after CTX. One possibility for this difference is that it takes time for the host (non-cancer) cells to respond to the stressor (CTX) fully, making the 4-day data more robust than the 2-day data. Another possibility is that there were still residual amount of CTX in mice at 2 days post-injection, killing the injected cancer cells and thus reducing the enhancement of cancer burden by CTX. This critique is similar to that from Reviewer #2 (Critique 2) and we added the reason for selecting day 4 in Section 2.1.
(b) Fig. 1b: The quick answer is both. The rapid depletion of cancer cells in bar 4 (CTX-treated KO mice) is due to both low extravasation and higher macrophage cytotoxicity.
(c) Fig. S2: The reason for the “double injection” experiment is to address a caveat of the pre-treatment model: it is not used in clinical practice. To address this caveat, we used the “double injection model,” where chemotherapy is administered in a clinically relevant context: neoadjuvant therapy—chemotherapy in the presence of primary tumor (as detailed in the manuscript). As shown in Fig. S2, chemotherapy also exacerbated cancer colonization in this model, lending credence to the pre-treatment model.
Despite mimicking a therapeutic modality in clinics, the double injection model has a critical drawback. Tumor sends out signals to affect the host, making it impossible to determine whether the change in lung environment is due to chemotherapy per se or chemotherapy plus signals from the primary tumor. The pre-treatment model does not suffer this drawback and is “cleaner.” Since the issue we sought to address is how chemotherapy exacerbates metastasis by affecting the non-cancer host cells, pre-treatment is the proper (correct) model to study the mechanisms.
In short, the reason for the “double injection” model is to test whether chemotherapy exacerbates metastasis in the context of a clinically relevant therapeutic modality—in order to complement the main model in our study. However, to avoid confounding factors, we used the pre-treatment model in the rest of the manuscript to study mechanisms. We clarified these points in the Result section, which entails moving some text from Discussion up to Section 2.1.
Critique 2: Fig. 2a - It may be confusing to the readers when this figure shows that cells have all extravasated within 36h while there are still colonies after 72h in Fig.1b.
Response: The colonies at 72 hr are from cancer cells that have already extravasated (thus, in the lung parenchyma). Colonies (1 or more cells, within 10 μm of each other as specified in Section 2.1) were counted regardless whether the cancer cells are inside the pulmonary blood vessels or outside (in the parenchyma).
Critique 3: Fig. 3b - Have the authors also tried to “add-back” KO donors-derived macrophages into WT recipients? Is the increased cytotoxic activity of the KO macrophages also cell-intrinsic, not needing the absence of Atf3 in the host cells?
Response: This is an interesting experiment. Although we have not done this exact experiment, our data from clodronate deletion using conditional knockout (CKO) mice addressed the cell-intrinsic issue as detailed in Section 2.5. We thank the reviewer for this suggestion.
Critique 4: Section 2.5 - I feel the authors have interpreted their data too rigidly at this point. Since “loss of Atf3” is not a natural state, it does not make sense to say “the loss of Atf3 have conferred increase cytotoxic activity to Atf3 KO macrophages” per se. It is better to reason that Atf3 acts as a negative regulator of immune surveillance; or indeed act to dampen the cytotoxic response as argued in the Discussion.
Response: Before the gene expression data, we believe that it is premature to bring up the idea that Atf3 is a dampener to suppress the cytotoxic response (an explanation at the molecular level). Therefore, we kept the interpretation of data—at this point of the report—in a generic manner: positive versus negative. In response to reviewer’s suggestion, we added the following sentence at the end of that paragraph to foreshadow the mechanistic explanation—at the molecular level.
“In sections 2.6 and 2.7 below, we will present data suggesting that the negative role of Atf3-/- macrophage (upon CTX treatment) is, at least in part, due to its lack of a dampener (Atf3) to repress cytotoxic genes that have cancer-killing potential (more in Discussion).”
Critique 5: Fig. 6 - The authors tried hard to elucidate the effectors of the increased cytotoxicity of Atf3 KO macrophages, but the candidate genes they highlighted are unexpected. Moreover, the absence of differential cytokine expression between the 4 subsets is inconsistent with the notion that Atf3 acts to “dampen cytokine storm” during sepsis. Is it possible that some of the samples have become exposed to bacterial contaminants or LPS and hence the induction of anti-microbial genes in Atf3 KO macrophages? Can the authors provide data that exclude this possibility?
Response: We address this comment in two parts.
(a) The lack of differential cytokine expression is likely due to the differences in the stress paradigms: CTX versus LPS. Thus, our data are not necessarily inconsistent with previous studies. In fact, the unexpected results are exciting, because they uncovered something new. We added this point in Discussion.
(b) We do not believe that the data (anti-microbial genes) are due to contamination for several reasons:
- We carried out all procedures (from tissue harvesting to cell sorting) under sterile conditions, and collected the sorted cells directly into TRIzol (inside a laminar hood) for RNA isolation.
- The differential expression of the anti-microbial genes are only observed between the CTX-treated groups (WT versus KO)—not between the PBS groups (WT versus KO). If contamination is the reason for us to detect the anti-microbial genes, the contamination would have to be selective to the CTX groups only, which is an unlikely event.
- The anti-microbial genes were validated by RT-qPCR in 5 independent experiments. If contamination is an issue, all these experiments had to be compromised—in a selective manner as indicated in (ii).
Taken together (i-iii), the likelihood of contamination is minimal. The fact that we did not see the cytokine storm—as in the LPS paradigm for bacterial infection—further argues against the contamination scenario.
Critique 6: Lastly, the authors should address the possibility that the loss of Atf3 alters the differentiation or polarisation of macrophages post-CTX treatment, using well established markers.
Response:
(a) Polarization: Previously, we analyzed the genes using the widely-known classic M1 and M2 markers but did not find obvious differential expression pattern. In response to reviewer’s comment, we explored the literature for more markers and found M2-skewing genes preferentially up-regulated by CTX in the WT macrophages. Note that many of them are novel (not the classic) M2 markers. We added the heatmap as Figure 6c and revised the text in section 2.6 accordingly.
(b) Differentiation: This is an issue that we are addressing in a separate project. Our preliminary data support the notion that CTX increases myeloid cell differentiation, as shown by its ability to increase the common myeloid progenitor cells (CMPs, which give rise to all myeloid-lineage of cells) and granulocyte-monocyte progenitors (GMPs, which differentiate into monocytes, macrophages and granulocytes) in the bone marrow. This was added as Figure S9d and described in Discussion.
Minor points:
(i) - p.3 line 118-120 and - p.3 line 124:
Response: We have edited the sentences as suggested by the reviewer:
(ii) Typographical errors:- p.3 line 104:
Response: We corrected the error.
REVIEWER #2:
Critique 1: Did authors examine the levels of Atf3 pre and post cyclophosphamide treatment at earlier time points …?
Response: This is an important question and we have been working on it for a long time. Theoretically, this is an easy question to address. However, it has been surprisingly difficult for us to detect a robust Atf3 induction by CTX in our model. We attribute our difficulties to at least 2 factors. (i) Due to its stress-inducible nature, Atf3 is induced by various stress factors during the isolation process (such as shear force, temperature fluctuation, etc.), increasing the basal level of ATF3 in the non-CTX group. (ii) Macrophage is a heterogeneous population of cells and the induction of Atf3 may only occur in sub-populations of macrophages. Together, these factors would reduce the signal to background ratio, making it difficult to detect a robust Atf3 induction by CTX.
In response to reviewer’s comment, we examined the early time points by co-immunofluorescence analysis of ATF3 (protein) and F4/80+ (a macrophage marker). Data in Figure S4c support the notion that Atf3 is induced by CTX at earlier time points. We also tested Atf3 induction at 16 hours post-CTX injection. Since this was before cancer cell injection, monocytes (macrophage precursors) were not recruited to the lung yet. Thus, we analyzed blood samples for Atf3 gene expression in myeloid cells (CD11b+). Figure S4d shows a trend of Atf3 induction, suggesting that CTX by itself—in the absence of cancer signal—can elicit response from the Atf3 gene, consistent with its nature as a stress gene. We added the data in supplemental and the description in Section 2.3.
Critique 2: Did authors do pilot studies to determine optimal window of CTX-pre treatment. Different time intervals were reported in the literature for CTX pre-treatment as the reported half life of CTX is 3-12hrs. Could authors please explain the basis to administer CTX four days prior to injection of cancer cells?
Response: As described in our response to Reviewer #1, Critique 1 (sub-point a), we have data on 2-day pre-treatment but the data are not as robust as those from 4-day pre-treatment (for potential explanations, see response above to Reviewer #1). We have no data on 3-day pre-treatment. We added the reason for selecting 4-day pre-treatment in Section 2.1.
Critique 3: The authors have reported "...increase in cancer cells by CTX was augmented at later time points in the WT lung, but dwindled away in the KO lung" but the data in Figure 1B shows fluctuating number of colonies/FOV rather than consistent augmented pattern across timepoints in WT pre-treated with CTX? Could authors please comment on it.
Response: This point is well-taken and we changed the word “augmented” to “maintained.”
Critique 4: The authors have reported less cancer cell burden as the plausible reason for decreased cell proliferation noticed at 9hr timepoint in KO mice treated with CTX in Fig S3. The data is reflective of percentage of p-HH3+ cells/total cells counted. In this context could authors please comment how it could be attributed to less cancer cell burden?
Response: We believe that this is a mis-reading of our statement, which meant the other way around. We stated in the manuscript “This exception can contribute to a lower cancer burden in this group.” That is, a lower proliferation can contribute to a lower cancer burden. We modified the statement by changing the word “can” to “may” to be more circumspect.
Critique 5: Could authors please … cite the reference correctly (example: The references (15, 16, 19) cited in Ln # 58 did not treat mice with cyclophosphamide in their studies).
Response: The reason for this comment is likely due to the fact that the CTX result was published in the supplemental material—not the main figures. In response to this comment, we cited the references separately (instead of lumping them together) and specifically referred to the supplemental material.
Minor Points:
(i) correct the format to denote the cell numbers:
Response: We corrected the error.
(ii) “fat pat to “fat pad”:
Response: We corrected the error.

Reviewer 2 Report
The authors have investigated the pattern of lung colonization in breast cancer and the mechanism using a pre-treatment model. The study is extension of authors previous findings implicating increased breast cancer cells colonization in lungs of WT mice but not in Atf3 KO mice. In this study the authors have shown Atf3 status could regulate extravasation and define the fate of macrophage activity either as anti-apoptotic in the WT or pro-apoptotic in the Atf3 KO macrophages upon treatment with CTX. The author have also reported increased antimicrobial genes with anti-cancer properties in Atf3 KO compared to WT macrophages. It is interesting study but the manuscript could not be considered in the current form for following reasons.
Major concerns:
1) Did authors examine the levels of Atf3 pre and post cyclophosphamide treatment at earlier time points (3h, 9h, 18h and 36h)? The authors have shown data for 72h but not for earlier timepoints. Could authors please include data for earlier timepoints especially 9h - 36h?
2) Did authors do pilot studies to determine optimal window of CTX-pre treatment. Different time intervals were reported in the literature for CTX pre-treatment as the reported half life of CTX is 3-12hrs. Could authors please explain the basis to administer CTX four days prior to injection of cancer cells?
3) The authors have reported "...increase in cancer cells by CTX was augmented at later time points in the WT lung, but dwindled away in the KO lung" but the data in Figure 1B shows fluctuating number of colonies/FOV rather than consistent augmented pattern across timepoints in WT pre-treated with CTX? Could authors please comment on it.
4) The authors have reported less cancer cell burden as the plausible reason for decreased cell proliferation noticed at 9hr timepoint in KO mice treated with CTX in Fig S3. The data is reflective of percentage of p-HH3+ cells/total cells counted. In this context could authors please comment how it could be attributed to less cancer cell burden?
5) Could authors please review the manuscript carefully and cite the reference correctly (example: The references (15, 16, 19) cited in Ln # 58 did not treat mice with cyclophosphamide in their studies).
Minor concerns:
1) Could authors please use correct format to denote the cell numbers (example: Please use superscript to denote the cell number across the manuscript).
2) Please replace "fat pat" with "fat pad" on Ln # 491 of pg # 13.
Author Response
REVIEWER #2:
Critique 1: Did authors examine the levels of Atf3 pre and post cyclophosphamide treatment at earlier time points …?
Response: This is an important question and we have been working on it for a long time. Theoretically, this is an easy question to address. However, it has been surprisingly difficult for us to detect a robust Atf3 induction by CTX in our model. We attribute our difficulties to at least 2 factors. (i) Due to its stress-inducible nature, Atf3 is induced by various stress factors during the isolation process (such as shear force, temperature fluctuation, etc.), increasing the basal level of ATF3 in the non-CTX group. (ii) Macrophage is a heterogeneous population of cells and the induction of Atf3 may only occur in sub-populations of macrophages. Together, these factors would reduce the signal to background ratio, making it difficult to detect a robust Atf3 induction by CTX.
In response to reviewer’s comment, we examined the early time points by co-immunofluorescence analysis of ATF3 (protein) and F4/80+ (a macrophage marker). Data in Figure S4c support the notion that Atf3 is induced by CTX at earlier time points. We also tested Atf3 induction at 16 hours post-CTX injection. Since this was before cancer cell injection, monocytes (macrophage precursors) were not recruited to the lung yet. Thus, we analyzed blood samples for Atf3 gene expression in myeloid cells (CD11b+). Figure S4d shows a trend of Atf3 induction, suggesting that CTX by itself—in the absence of cancer signal—can elicit response from the Atf3 gene, consistent with its nature as a stress gene. We added the data in supplemental and the description in Section 2.3.
Critique 2: Did authors do pilot studies to determine optimal window of CTX-pre treatment. Different time intervals were reported in the literature for CTX pre-treatment as the reported half life of CTX is 3-12hrs. Could authors please explain the basis to administer CTX four days prior to injection of cancer cells?
Response: As described in our response to Reviewer #1, Critique 1 (sub-point a), we have data on 2-day pre-treatment but the data are not as robust as those from 4-day pre-treatment (for potential explanations, see response above to Reviewer #1). We have no data on 3-day pre-treatment. We added the reason for selecting 4-day pre-treatment in Section 2.1.
Critique 3: The authors have reported "...increase in cancer cells by CTX was augmented at later time points in the WT lung, but dwindled away in the KO lung" but the data in Figure 1B shows fluctuating number of colonies/FOV rather than consistent augmented pattern across timepoints in WT pre-treated with CTX? Could authors please comment on it.
Response: This point is well-taken and we changed the word “augmented” to “maintained.”
Critique 4: The authors have reported less cancer cell burden as the plausible reason for decreased cell proliferation noticed at 9hr timepoint in KO mice treated with CTX in Fig S3. The data is reflective of percentage of p-HH3+ cells/total cells counted. In this context could authors please comment how it could be attributed to less cancer cell burden?
Response: We believe that this is a mis-reading of our statement, which meant the other way around. We stated in the manuscript “This exception can contribute to a lower cancer burden in this group.” That is, a lower proliferation can contribute to a lower cancer burden. We modified the statement by changing the word “can” to “may” to be more circumspect.
Critique 5: Could authors please … cite the reference correctly (example: The references (15, 16, 19) cited in Ln # 58 did not treat mice with cyclophosphamide in their studies).
Response: The reason for this comment is likely due to the fact that the CTX result was published in the supplemental material—not the main figures. In response to this comment, we cited the references separately (instead of lumping them together) and specifically referred to the supplemental material.
Minor Points:
(i) correct the format to denote the cell numbers:
Response: We corrected the error.
(ii) “fat pat to “fat pad”:
Response: We corrected the error.

Round 2
Reviewer 1 Report
The authors have adequately addressed my comments. I thank them for their careful explanation and clear articulation of their reasons.
Reviewer 2 Report
The authors have addressed the comments and the manuscript could be considered for publication.